# Microstructure and Mechanical Properties of AA7075 Aluminum Alloy Fabricated by Spark Plasma Sintering (SPS)

**DOI:** 10.3390/ma14020430

**Published:** 2021-01-16

**Authors:** Elder Soares, Nadège Bouchonneau, Elizeth Alves, Kleber Alves, Oscar Araújo Filho, David Mesguich, Geoffroy Chevallier, Christophe Laurent, Claude Estournès

**Affiliations:** 1Mechanical Engineering Department, Federal University of Pernambuco (UFPE), Av. da Arquitetura, S/N, Recife, PE 50.740-550, Brazil; elder.soares52@gmail.com (E.S.); nadege.bouchonneau@gmail.com (N.B.); elizeth_oliveira@yahoo.com.br (E.A.); kleber.gbalves@ufpe.br (K.A.); oscaroaf98@hotmail.com (O.A.F.); 2CIRIMAT, Université de Toulouse, CNRS, Université Paul-Sabatier, 118 Route de Narbonne, 31062 Toulouse CEDEX 9, France; mesguich@chimie.ups-tlse.fr (D.M.); chevallier@chimie.ups-tlse.fr (G.C.); laurent@chimie.ups-tlse.fr (C.L.)

**Keywords:** Spark Plasma Sintering (SPS), aluminum alloy, microstructure, mechanical properties

## Abstract

AA7075 aluminum alloy is widely used for several high-technology applications for its high mechanical strength to weight ratio but is still the subject of several studies seeking a further increase in its mechanical properties. A commercial powder is used, either as-received or after ball-milling. Dense AA7075 samples are prepared in one step by Spark Plasma Sintering, at 550 °C with a holding time of 15 min and a uniaxial pressure of 100 MPa. No additional heat treatment is performed. Laser granulometry, X-ray diffraction and optical- and scanning electron microscopy show that both grain size and morphology are preserved in the dense samples, due to the relatively low temperature and short sintering time used. The samples prepared using the ball-milled powder exhibit both higher Vickers microhardness and transverse fracture strength values than those prepared using the raw powder, reflecting the finer microstructure.

## 1. Introduction 

Powder metallurgy (PM) technology has been of interest to several industrial sectors for decades. The necessity for metallic materials with improved mechanical and physical properties to satisfy the current requirements has resulted in development of new manufacturing processes [1,2]. The AA7075 aluminum alloy is widely used in the aeronautical, marine, automotive and defense industries for its high mechanical strength to weight ratio [3,4], but is still the subject of several studies seeking to further improve its mechanical properties [5].

The manufacture of aluminum alloy parts commonly uses powder metallurgy processes, such as cold isostatic pressing (CIP), hot isostatic pressing (HIP), hot pressing (HP), hot extrusion and hot rolling. These are competitive with respect to casting and extrusion, but it involves the degassing of the powder for several hours and several more hours (1–5 h) at relatively high temperatures [6,7], which makes the process expensive [1,2,8]. Moreover, the long cycle time and high temperatures lead to the growth of the grains, to the detriment of the final properties.

Recent research has enabled the development of a new unconventional sintering technology that has emerged as one of the best techniques for consolidating metallic powders. Although several nomenclatures are used to describe the method, including pulsed electric current sintering (PECS), field assisted sintering (FAST), field-activated sintering technique, and current-activated pressure-assisted densification, it is often named Spark Plasma Sintering (SPS) [1,9,10]. During the last decade, the SPS technique has been used to consolidate a wide variety of materials, producing them with high density in a short processing time [11,12,13,14].

The SPS technique involves applying a pulsed direct current simultaneously with uniaxial pressure, which allows one to reach high heating rates, performing sintering at relatively low temperatures for short sintering times and enabling the preparation of highly dense materials without the use of pre-compaction, binders (and degassing), in addition to providing materials with better properties according to the fine grain size of the matrix [2,15,16,17,18,19]. The success of the SPS is attributed, according to M. Tokita [20], to the role of the spark and plasma (confer the name of the technique) that are generated between the particles [21,22]. Although the concept of plasma is plausible, occurrence is not a consensus [23], and the role of pulsed current is not properly understood [15].

Another advantage is that SPS is also suitable for pure or pre-alloyed aluminum powders because it is a highly effective method to break the surface film of Al_2_O_3_-3H_2_O, that represents a major impediment to conventional sintering in the solid state [1].

In general, SPS has the potential to emerge as a metal consolidating technology that can change the way certain aerospace components are manufactured. However, this must first be demonstrated with aluminum alloys of direct relevance to the aerospace sector [1], such as the AA7075 aluminum alloy. Therefore, the aim of this study is to investigate the microstructure and mechanical properties of AA7075 aluminum alloy samples produced by mechanical alloying followed by SPS.

## 2. Materials and Methods 

The AA7075 powder (ALCOA, Brazil [24]) has an average particle size of 30 μm. The chemical composition of the as-received AA7075 powder determined from EDX analysis is presented in Table 1. The powder was used both as-received and after ball-milling in a high-energy mill (SPEX), consisting in a 304 L stainless steel jar and SAE 52100 steel balls approximately 6.2 mm in diameter. The powder was ball-milled in isopropyl alcohol for 1 h with 2 wt.% stearic acid (CH_3_(CH_2_)_16_COOH). Milling was performed with a ball to powder weight ratio of 10:1 [7,25]. The raw and ball-milled powders will be referred to hereafter as AA7075 and AA7075-M, respectively.

The powders were consolidated by Spark Plasma Sintering (Dr Sinter SPS 632Lx, Fuji Electronic Industrial CO., Saitama, Japan) at the Plateforme Nationale CNRS de Frittage Flash (Toulouse, France). The powders were loaded into a graphite die. A sheet of graphitic paper was placed between the punch and the powder and between the die and the powder for easy removal. The SPS process was performed in vacuum (residual cell pressure < 10 Pa) using a direct current pulse pattern of 40 ms: 7 ms (pulse on: pulse off). The temperature was controlled using a K-thermocouple introduced in a hole (5 mm deep) drilled on the outer surface of the die. The samples were heated (100 °C/min) up to 550 °C where a 15 min dwell was applied. A uniaxial charge (corresponding to 100 MPa on the powder bed) was applied at room temperature and maintained during the heating and dwell steps (Figure 1). Natural cooling was applied down to room temperature and the uniaxial load was gradually released at the same time.

The SPS pellets were 20 mm in diameter and 3 mm thick. The samples obtained from the AA7075 and AA7075-M powders are referred to hereafter as AA7075-S and AA7075-MS, respectively. The graphitic paper remaining on the surface was removed by machining. Initially, the grinding of the samples was performed with SiC papers (grades 1200 and 2400). Then, the samples were polished in four steps: 6, 3, and 1 µm diamond suspensions and 20 nm amorphous silica suspension [26]. Etching (10 g NaOH + 100 mL H_2_O [27] during 10 s) was performed at room temperature in order to reveal the grain boundaries. To stop the etching process, the samples were immersed in ethanol for 20 s and then immersed in distilled water for 1 min under ultrasound cleaning. The relative densities were obtained by the Archimedes method (average value of 10 measurements).

The crystalline phases in the powders and sintered samples were detected and identified using X-ray diffraction (XRD, Bruker D4 (Cu K_α_) in *θ*–2*θ* configuration), in the range of 10° < 2*θ* < 100° with a step size of 0.02°, using CuK_α1_ radiation (0.15406 nm), having the accelerating voltage of 40 kV. The crystallite size and the micro-strain have been estimated from the XRD patterns using the Williamson–Hall (W–H) method [28]. The XRD peak measured broadening normally consists of two parts, which are the physical and instrumental broadening. The corrected physical broadening (β) can be obtained using Equation (1):(1)$$\beta^{2}=\beta_{m}^{2}-\beta_{i}^{2},$$
where $\beta_{m}$ is the experimental full-width at half maximum (FWHM) and $\beta_{i}$ is the instrumental broadening obtained by Caglioti equation [29].

After subtracting the instrumental broadening, the corrected physical broadening (β) can be considered as the sum of widths due to small crystallite sizes ($\beta_{D}$) and lattice strains ($\beta_{\varepsilon }$) [30] as shown by Equation (2):(2)$$\beta =\beta_{D}+\beta_{\varepsilon },$$

The peak broadening due to the crystallite size *β**_D_* is expressed by Equation (3) [28]:(3)$$\beta_{D}=\frac{K\lambda }{D\cos\theta },$$
where, *K* is a constant (*K* = 0.94 for small cubic crystal [31]), *λ* is the wavelength of the X-rays used, *D* is the crystallite size and *θ* is the Bragg angle.

Similarly, the peak width due to lattice strain $\beta_{\varepsilon }$ is (Equation (4):(4)$$\beta_{\varepsilon }=4\varepsilon \tan\theta ,$$

From Equations (3) and (4), we get
(5)$$\beta =\frac{K\lambda }{D\cos\theta }+4\varepsilon \tan\theta ,$$
or,
(6)$$\beta \cos\theta =\varepsilon \left(4\sin\theta \right)+\frac{K\lambda }{D},$$

Equation (6), above, is an equation of a straight line of the type y=ax+b, where the slope a of this straight line provides the value of the intrinsic strain ε and the intercept b gives the average particle size D, so ε=a and D=Kλ/b. 

The micro-strain *ε* and the crystallite size *D* obtained by XRD analyses were used to calculate the dislocation density *δ*, expressed by the Equation (7) [32].
(7)$$\delta =2\sqrt{3}{\langle \varepsilon^{2}\rangle }^{1/2}/\left(D\times b\right),$$
where *δ* is the dislocation density and b is the magnitude of the Burgers vector equal to $a_{0}/\sqrt{2}$ for an FCC Al alloy.

The particle-size distribution (PSD) of the AA7075 and AA7075-M powders was obtained by laser granulometry (Malvern Instruments Mastersize 2000). The samples were observed by optical microscopy (Keyence VHX-1000 Digital Optical Microscope) and field-emission-gun scanning electron microscopy (FESEM, JEOL 6400). The density of sintered samples was measured by Archimedes′ principle (Sartorius YDK01). The indentation tests (100 g (0.98 N) for 10 s in air at room temperature) were performed on the polished surface of the specimens by loading with a Vickers indenter (Mitutoyo HM 2000). The values reported are the average of 10 measurements. The transverse rupture strength (σ_u_) was measured, parallel to the SPS pressing axis, by the three-point bending method (Material Testing Systems MTS 1/M) on specimens about 2 × 2 × 16 mm^3^. The span between the two supporting pins is equal to 13 mm. Crosshead speed was fixed at 0.1 mm/min. 

## 3. Results and Discussion

Typical FESEM images of the AA7075 powder at medium (Figure 2a) and higher (Figure 2b) magnifications show rounded particles with a spherical or more elongated shape and a large size distribution, consistent with the gas atomization practices employed for its production [33]. For the AA7075-M powder, the particles have taken the shape of micrometric flakes about 1 µm thick and with lateral dimensions in the range 10–100 µm, which is a typical morphology for milled soft metals for a short time (≤ 5 h), as reported in previous works [34,35]. In particular, Razavi-Tousi and Szpunar [35] found that using various milling conditions, aluminum particles always tend to form disc-shaped particles for short milling times due to cold welding. For longer times (10–50 h), the particle size and its final shape are determined by the equilibrium between cold welding and fracture processes.

The particle-size distribution (PSD) is shown in Figure 3. The median particle size (D_50_) is equal to 33 µm for AA7075 (Figure 3a) and to 56 µm for AA7075-M (Figure 3b). The increase in particle size after 1 h of grinding in an aluminum alloy (AA6061) was also observed by Rana et al. [36]. They showed that the particle size increased from 10–12 µm to 60–65 µm, and attributed this increase to cold welding between the particles, which results in the formation of agglomerates in which the particles are weakly joined at the point of contact. Both PSD are large, in agreement with the SEM observations, and interestingly the PSD for the AA7075-M powder is bimodal, with the minor component centered at about 500 µm, which could also reflect some effects of ball-milling, such as flaking and inter-particle bonding.

The XRD patterns of the AA7075 and AA7075-M powders and the corresponding sintered samples are shown in Figure 4a. For all samples, only the aluminum peaks are detected (space group n 225: $Fm\bar{3}m$; Schoenflies notation: $O_{h}^{5}$), showing that all other crystallized phases formed by the other alloying elements are below the detection limit of the technique [37]. Figure 4b shows the magnification of the region around the (111) peak. The (111) peak shifts to a slightly higher 2*θ* angle, seen in Figure 4b for sintered samples, which can be attributed to the partial dissolution of Zn, Mg and Cu in the Al matrix as well as grain boundaries [38]. The values of crystallite size and intrinsic strain for all samples are shown in Table 2. To obtain these parameters, the values of FWHM and Bragg angle of the first five peaks of the diffraction pattern, indexed as (111), (200), (220), (311) and (222), were used. The high-energy ball milling process led to a decrease in crystallite size and an increase in lattice strain, due to the severe plastic deformation of the powders during the process, as was reported by several authors [32,39]. The plastic deformation generates stresses that imply the formation of subgrain boundaries, thus, grain refinement occurs as the distortions in the lattice increase. The crystallite size is equal to about 61 and 45 nm and strain is about 0.07 and 0.12%, for AA7075 and AA7075-M, respectively. Table 2 also shows the estimated value of the density of dislocations for each sample, which was obtained using Equation (7). Comparing values for AA7075 and AA7075M highlights that milling considerably increases the density of dislocations [34,40]. Moreover, sintering reduces the density of dislocations, as it increases the size of the crystallite and decreases the strain of the lattice. In addition, comparing AA7075-MS with AA7075-S, it is possible to notice that the density of dislocations for the former is about four times higher than for the latter, which influences directly the strength and hardness of these samples, as will be discussed hereafter.

Using Bragg′s Law for cubic structures, the lattice parameter *a*_0_ of the powder and consolidated samples was calculated. Comparing the lattice parameter of the powder samples (0.40549 nm for AA7075 and 0.40553 nm for AA7075M) shows that there is no significant variation. The same occurs for AA7075-S and AA7075-MS, with lattice parameters of 0.40498 and 0.40491 nm, respectively. Comparing the consolidated samples with the powder samples, there is a reduction of about 0.13–0.15%. The reduction in the *a*_0_ occurs due to the formation of a supersaturated AA7075 solid solution caused by the dissolution of smaller Zn and Cu atoms in the Al lattice [38,41].

The relative density of the AA7075-S and AA7075-MS sintered samples is equal to 99.2 ± 0.1% and 99.3 ± 0.2%, respectively, using 2.81 g/cm^3^ for the theoretical density of AA7075. The XRD patterns of the sintered samples (Figure 4) are similar to those of the corresponding powders.

From the Williamson–Hall method, it was found that the crystallite size of the sample AA7075-S was 80 nm, about 32% larger than its starting powder, its micro-strain was reduced from 0.07% to 0.02% after sintering. The crystallite size and micro-strain in the sintered AA7075-MS sample are 77 nm and 0.08%, respectively. The crystallite size of this sintered sample is approximately 70% larger than that of its powder. Such results are in good agreement with those obtained by Rana et al. [36].

The microstructure of the sintered samples was observed by optical microscopy on cross-sections made perpendicular to the SPS pressing direction. The images (Figure 5) reveal only very little or no porosity, in agreement with the relative densities higher than 99%. For AA7075-S (Figure 5a), the grain shape is mostly isotropic, reflecting the shape of the particles in the corresponding powder. The average grain size was estimated from the image by the ratio of the lengths of four randomly drawn lines to the amount of grain outlines that intersect them. It is equal to 34 ± 2 µm, which is close to the median value for the corresponding powder (33 µm), indicating no or very little grain growth during SPS. The AA7075-MS sample (Figure 5b) exhibits a lamellar morphology, with lamellae a few micrometers thick and dozens of micrometers in length, as for the starting powder. Some grains are observed on the thin side and some on the broad side, indicating no preferential orientation with respect to the SPS pressing axis.

The results of the mechanical tests are presented in Table 3. Both the Vickers microhardness H (HV) and transverse rupture strength σ_u_ (MPa) are higher for AA7075-MS than for AA7075-S, whereas the transverse rupture strain ε_u_ is lower. These results could reflect the changes in crystallite size, micro-strain and dislocation density caused by ball-milling for the AA7075-M powder, which are associated with both dislocation and solid solution strengthening. It is well known that metals sintered from milled powders tend to have higher hardness and strength than powders sintered as received due to grain refinement; however, they tend to become less ductile [42,43]. Increasing the strength of metals without major losses of ductility still remains a challenge in materials engineering [44]. Post-heat treatments, such as annealing, are used to regain ductility [36,45], reducing the residual stress of dense samples. The residual stress in materials is undesirable as it can cause pre-existing cracks that result in material damage or early fracture [46]. Research on methods to recover the ductility without reducing the strength of dense metallic samples is of great interest for further works. Higher dislocation density makes plastic deformation more difficult, as dislocations interact with each other and act as barriers that hinder their own movement. Thus, the increase in the dislocation density in a metal increases the yield strength of the material [47]. The solid solution strengthening occurs due to the presence of other elements (such as Zn, Mg, Cu) alloyed with the Al matrix as solute atoms that differ from the matrix atoms in size, which can cause a variation in the strain fields. These created local strain fields interact with the dislocations and prevent their movement, also causing an increase in the yield strength of the material [48]. Therefore, these two strengthening mechanisms are associated with each other, as the greater the dislocation density, the greater their interaction with the local strain fields created by the elements in solid solution.

It is interesting to underline that the values for the present samples prepared by the one-step SPS technique are higher than those reported for samples prepared by conventional manufacturing processes using long periods of heat treatment. For example, AA7075 samples prepared by casting and T6 thermal treatment [50] showed a bending strength equal to about 330 MPa. Using a vertical squeeze casting process, Kalkanli and Yilmaz [49] obtained AA7075 samples with a bending strength equal to about 400 MPa (as-cast) and 450 MPa (T6).

## 4. Conclusions

AA7075 aluminum alloy powders used as received or after ball-milling were consolidated into dense (>99%) samples by SPS. It is shown that the relatively low temperature and short sintering time do not cause appreciable grain growth and thus preserves the grain size and morphology of the powders. The XRD analysis highlights the (111) peak shift to a slightly higher 2*θ* angle for sintered samples, which can be attributed to the partial dissolution of Zn, Mg and Cu in the Al matrix, as well as grain boundaries. The sample prepared using the ball-milled powder (AA7075-MS) shows a higher Vickers microhardness and transverse rupture strength (>500 MPa) than the one prepared using the raw powder (AA7075-S), reflecting the finer microstructure. Comparing AA7075-MS with AA7075-S, it is possible to notice that the density of dislocations for the former is about four times higher than for the latter, which influences directly the strength and hardness of these samples. Higher dislocation density makes plastic deformation more difficult, as dislocations interact with each other and act as barriers that hinder their own movement. Thus, the increase in the dislocation density in a metal increases the yield strength of the material. Importantly, such values obtained for the present samples are higher than those obtained for some samples prepared by routes involving additional heat treatments. The potential of SPS as an economical and fast option for the manufacture of AA7075 aluminum alloy products with superior mechanical properties will be explored into more details in future works. In further studies, it would be interesting to analyze the effect of the milling time on the crystallite size, micro-strain, dislocation density, and on the microstructure and mechanical properties of the AA7075 and/or other aluminum alloys sintered by SPS. In order to optimize and improve the material properties, it would be interesting to investigate the influence of natural (room temperature) and artificial (T6, among others) aging treatments in relation to the hardness and mechanical resistance in the final product.

## Figures and Tables

**Figure 1 materials-14-00430-f001:**
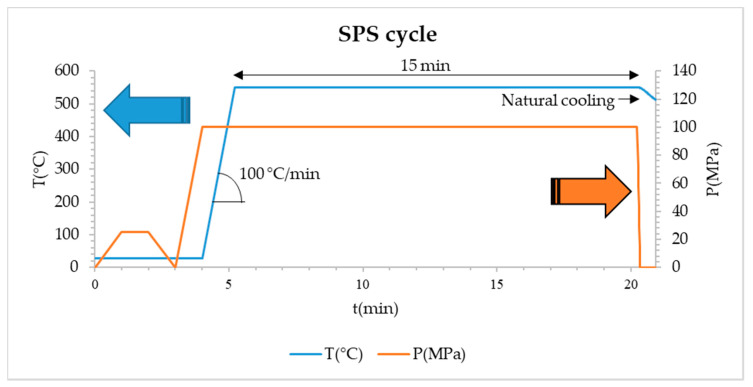
SPS cycle. The blue line represents the temperature cycle, while the orange line represents the pressure cycle.

**Figure 2 materials-14-00430-f002:**
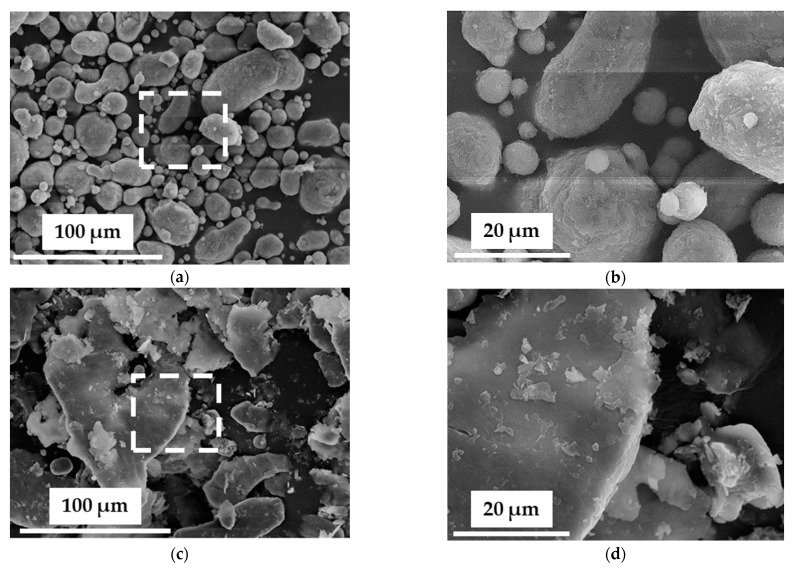
FESEM images of the powders: (**a**) AA7075 (as-received powder); (**b**) higher magnification of the boxed area in (**a**); (**c**) AA7075-M (milled powder); (**d**) higher magnification of the boxed area in (**c**).

**Figure 3 materials-14-00430-f003:**
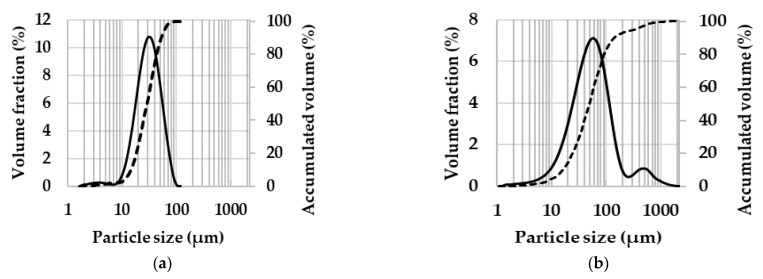
Particle size distribution: (**a**) AA7075 powder; (**b**) AA7075-M powder. The solid lines represent the volume fraction and the dashed lines represent the cumulative volume.

**Figure 4 materials-14-00430-f004:**
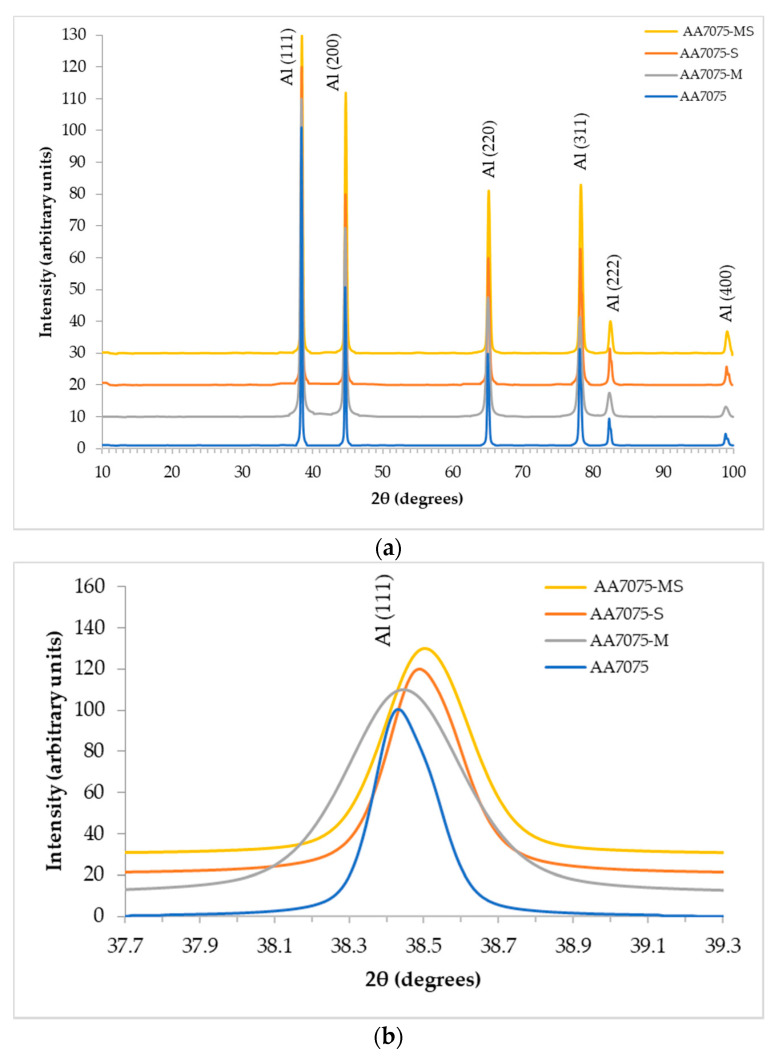
(**a**) XRD patterns of the AA7075 and AA7075-M powders and the corresponding sintered samples; (**b**) higher magnification of the region around the (111) peak. All patterns are normalized to the (111) peak.

**Figure 5 materials-14-00430-f005:**
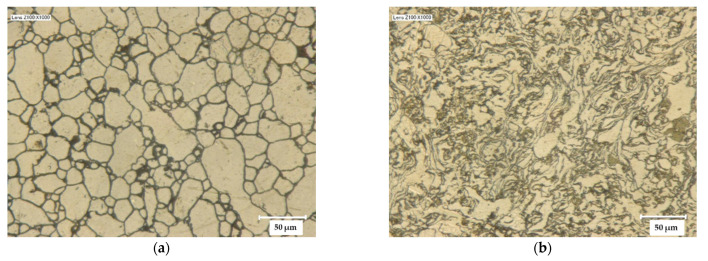
Optical images of the sintered samples: (**a**) AA7075-S, (**b**) AA7075-MS.

**Table 1 materials-14-00430-t001:** Chemical composition of the AA7075 raw powder determined from EDX analysis.

	Si	Cu	Cr	Mg	Zn	Al
wt.%	0.10	1.38	0.27	1.71	5.06	91.48

**Table 2 materials-14-00430-t002:** Crystallite size D (nm), micro-strain ε (%), lattice parameter a_0_ (nm) and dislocation density δ (m^−2^) powders and dense samples as deduced from the XRD patterns.

Sample	D (nm)	ε (%)	a_0_ (nm)	δ (m^−2^)
AA7075	61	0.07	0.40549	1.37 × 10^14^
AA7075-M	45	0.12	0.40553	3.26 × 10^14^
AA7075-S	80	0.02	0.40498	2.96 × 10^13^
AA7075-MS	77	0.08	0.40491	1.32 × 10^14^

**Table 3 materials-14-00430-t003:** Mechanical properties of the AA7075-S and AA7075-MS samples: Vickers microhardness H (HV), transverse rupture strength σ_u_ (MPa), transverse rupture strain ε_u_ (—).

Sample	H (HV)	σ_u_ (MPa)	ε_u_ (—)
AA7075-S	86 ± 2	498 ± 12	(5.9 ± 0.4)·10^−2^
AA7075-MS	108 ± 2	550 ± 5	(3.2 ± 0.1)·10^−2^
AA7075-AC [49]	—	≅ 400	—
AA7075-T6 [49]	—	≅ 450	—

## Data Availability

Data sharing not applicable.

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
