# Peer review of "Microstructure and Mechanical Properties of AA7075 Aluminum Alloy Fabricated by Spark Plasma Sintering (SPS)"

_materials, 2021, doi:10.3390/ma14020430_

Round 1

Reviewer 1 Report

The manuscript could be published in the present form.

Author Response

Response: Authors thank very much the reviewer for its report and publication recommendation.

Reviewer 2 Report

Dear authors,

thank you very much for submission of the very interesting and helpful article. I recommend to publish the article after minor revisions. Please, try to incorporate my few remarks to the manuscript.

Line 23 - please, use a full stop after the sentence instead of comma.

Line 88 - please, use an indefinite article “An” in the phrase “A uniaxial”.

In Line 145 based in previous work, authors stated that “For the AA7075-M powder, the particles have taken the shape of micrometric flakes about 1 µm thick and with lateral dimensions in the range 10-100 µm, which is a typical morphology for milled soft metals for a short time”. Could authors specify the duration of such a short time milling, approximately? And what is usually duration of long time milling, approximately as well (Line 147)?

Lines 149 and 150 - there should not be an article “a” before a word “magnification” in the description of Figure 2.

Line 149 - please, add values of magnifications to the description of the Figure 2.

Line 149 - in the description of the Figure 2, namely “FESEM images for the powders” is grammatically correct to use a preposition “of” instead of “from”.

Line 222 - Vickers microhardness is designated in capital letters HV.

Best regards

Author Response

Dear authors,

Thank you very much for submission of the very interesting and helpful article. I recommend to publish the article after minor revisions. Please, try to incorporate my few remarks to the manuscript.

  1. Line 23 - please, use a full stop after the sentence instead of comma.

Response: The comma has been replaced by the long term on line 23.

low temperature and short sintering time used. The samples prepared using the ball-milled powder

  1. Line 88 - please, use an indefinite article “An” in the phrase “A uniaxial”.

Response: The indefinite article was corrected in the sentence on line 88.

550 °C where a 15 min dwell was applied. An uniaxial charge (corresponding to 100 MPa on the

  1. In Line 145 based in previous work, authors stated that “For the AA7075-M powder, the particles have taken the shape of micrometric flakes about 1 µm thick and with lateral dimensions in the range 10-100 µm, which is a typical morphology for milled soft metals for a short time”. Could authors specify the duration of such a short time milling, approximately? And what is usually duration of long time milling, approximately as well (Line 147)?

Response: The time that the particles remain in the form of flakes will depend a lot on the milling conditions, remaining in this format for up to about 5 hours, being considered a short milling period. We can consider a long milling time around 10 to 50 hours.

for a short time (≤ 5h), as reported in previous works [35,36]. In particular, Razavi-Tousi and Szpunar [36] found that using various milling conditions, aluminum particles always tend to form disc-shaped particles for short milling times due to cold welding. For longer times (10 – 50h), the particle size and

  1. Lines 149 and 150 - there should not be an article “a” before a word “magnification” in the description of Figure 2.

Response: Article "a" has been removed.

Figure 2. FESEM images for the powders: (a) AA7075 (as-received powder); (b) higher magnification of the boxed area in (a); (c) AA7075-M (milled powder); (d) higher magnification of the boxed area in (c).

  1. Line 149 - please, add values of magnifications to the description of the Figure 2.

Response: Authors do not agree with the reviewer's comment. As the original scale is present in the figure we consider that the magnification is not necessary. Indeed, as the size of the image can be changed from its initial value at any time (i.e. during writing or editing the paper etc...) as a consequence we consider that indicating a magnification value does not make sense.

  1. Line 149 - in the description of the Figure 2, namely “FESEM images for the powders” is grammatically correct to use a preposition “of” instead of “from”.

Response: In the sentence "FESEM images for the powders", "for" was replaced by "of" on line 149.

Figure 2. FESEM images of the powders: (a) AA7075 (as-received powder); (b) higher magnification

  1. Line 222 - Vickers microhardness is designated in capital letters HV.

Response: HV was written in capital letters.

(HV) and transverse rupture strength σ(MPa) are higher for AA7075-MS than for AA7075-S,

Reviewer 3 Report

The authors made extensive and satisfactory corrections.

Author Response

Response: Authors thank very much the reviewer for its report and publication recommendation.

This manuscript is a resubmission of an earlier submission. The following is a list of the peer review reports and author responses from that submission.

Round 1

Reviewer 1 Report

The work discussed the subject Microstructure and Mechanical Properties of AA7075 Aluminum Alloy fabricated by Spark Plasma Sintering (SPS). Presented research stays interesting, in the reviewer opinion, however, at the present form remain close to a technical report nether research article. The quality of work should be improved before publishing. At the moment I advise against the publishing.

General

The general problem of this article lays in the scanty number of the analysis planed to characterized the majority of the applied approach. The superficiality of the analysis gives a response in the mechanical tests however remains unconnected with feedstock characteristics. The treatment procedures remain assumed, without any comments, literature reference or preliminary research results. The Sherrer formula can't be used for the powders with a strain effect. It uses after SPS processing and afterwards conclusions, remain also a serious flow of the work. Obtained results remain in dominance obvious and mostly remain in a lack of proper judgment.

Reviewer 2 Report

Dear authors,

thank you very much for the possibility to read about this highly interesting topic. But in this form, I have to reject your paper.

My comments are as follows, you can use it for the future research:

1. In Abstract (page 1, lines 20 to 22) there is only written, which techniques were used for analyzing the samples. It looks like “Experimental” part in scientific paper. I am missing describing some important results you gained during analyses.

2. In section “Materials and Methods” (page 2, line 49) is given the chemical composition provided bz manufacturer. It will be better to carry out some analysis to determine the exact chemical composition of powders.

3. Please, add the information on milling time ? (page 2, line 55) Did you investigate the influence of milling time on powder size?

4. Authors wrote, that 12 current pulses on and 2 current pulses off were used. What is the duration of such a pulse ? (page 2, line 64)

5. The etching was performed at the ambient temperature or at elevated temperature? (page 3, line 73). Please, add the information.

6. In Fig. 1a and 1b (page 3, line 102), I am missing the exact values of arbitrary units on the y-axis.

7. Please, add to the Fig. 1 except the planes, also phase being detected by XRD.

8. In x-axis of the Fig. 1, the decimals are designed with strokes, please rewrite it to dots. Furthermore, I am missing the XRD patterns of the sintered samples in Fig. 1 b.

9. In whole Results I am missing explanation of reached results. It is only some rough description of reached results, but no deep insight from the materials science point of view is given. For the future research, I recommend much deeper results – theoretical explanation relation.

10. There is 26 references, but 22 of them are given in Introduction and Experimental. It is necessary to include more research works to results for confrontation with the reached results.

Reviewer 3 Report

In introduction, I missed more detailed overview of results achieved by authors dealing with the same topic of SPS of AA7075 aluminium alloy. What was the method applied by the powder provider for measurement of the powder chemical composition? I recommend to measure chemical composition of milled powder before sintering and to compare the results with the chemical composition of powder given by the powder producer. I recommend to use “SPS process” instead of “SPS run” (line 63). I recommend to use “default machine parameters” instead of “default pattern of the machine” (line 64). For better visualization, I missed the time temperature diagram of SPS of the powders (heating rate, holding time) in the part “2. Materials and methods”. Please, in the case of current pulse explain values “12 ON and 2 OFF” (line 64). What is the current pulse duration? The statement in the sentence “The samples were polished in 6 steps: grade 1200 and 2400 - SiC paper; 6, 3, and 1 μm diamond suspension; and 20 nm amorphous silica suspension” is not correct, because the first two steps are considered as grinding steps (line 72). Volume units are not given in mL, but in ml (line 73). I recommend to use the term “ultrasound cleaning” instead of “sonication” (line 75). I recommend to use the term “crystalline phases” instead of “crystallized phases” (line 77). The term “field-effect-gun” is not correct (line 81). It is better to use the term “field emission gun”. Maybe, it could be a good idea to add the scheme of SPS process to the chapter 2. In the chapter 3, please add information about phase detected with XRD, f.e. number of space group, Schoenflies notation (line 91). If it is possible, I recommend to use XRD for measurement of residual stresses of as-received and ball-milled powders as well. I recommend to substitute “the peaks are slightly thinner” (line 124) by “FWHM of diffraction peaks slightly decreased”. In table 3, I recommend to substitute the unit of transverse rupture strain “mm/mm” by dimensionless unit “-“. In chapter 3, I missed more detailed comparison with results achieved by another authors. I recommend to improve it.

Reviewer 4 Report

In this interesting work one phrase should be clarified.

In the Introduction, line 35-36 you wrote: “The manufacture of aluminum alloy parts by conventional powder metallurgy (P/M) processes such as hot pressing and hot isostatic pressing ….”

For my opinion the conventional P/M process is composed from two independent stages: pressing (densification) and sintering (consolidation), whereas hot pressing can be classified to unconventional process, where the pressing and sintering stages are combined into the one process. Could you please rewrite your sentence?